# HERMES: Human-to-Robot Embodied Learning from Multi-SouRce Motion Data for MobilE DexterouS Manipulation

**Zhecheng Yuan**[1,2*], **Tianming Wei**[1,2*], **Langzhe Gu**[1,2*], **Pu Hua**[1,2],
**Tianhai Liang**[1,2], **Yunapei Chen**[3], **Huazhe Xu**[1,2]

[1] Tsinghua University, [2] Shanghai Qi Zhi Institute, [3] Peking University

**Abstract:** Leveraging human motion data to impart robots with versatile manipulation skills has emerged as a promising paradigm in robotic manipulation. Nevertheless, translating multi-source human hand motions into feasible robot behaviors remains challenging, particularly for robots equipped with multi-fingered dexterous hands characterized by complex, high-dimensional action spaces. In this paper, we introduce HERMES, a human-to-robot learning framework for mobile bimanual dexterous manipulation. First, HERMES formulates a unified reinforcement learning approach capable of seamlessly transforming heterogeneous human hand motions from multiple sources into physically plausible robotic behaviors. Subsequently, to mitigate the sim2real gap, we devise an end-to-end, depth image-based sim2real transfer method for improved generalization to real-world scenarios. Furthermore, to enable autonomous operation in varied and unstructured environments, we augment the navigation foundation model with a closed-loop Perspective-n-Point (PnP) localization mechanism, ensuring precise alignment of visual goals and effectively bridging autonomous navigation and dexterous manipulation. Extensive experimental results demonstrate that HERMES consistently exhibits generalizable behaviors across diverse, in-the-wild scenarios, successfully performing numerous complex mobile bimanual dexterous manipulation tasks. Project Page https://hermes-manipulation.github.io/

## 1 Introduction

Humans continuously generate diverse bimanual manipulation data, inherently serving as natural guidance for robots to emulate human-like behaviors. Several previous studies [1, 2, 3, 4, 5] have attempted to extract trajectories of human hands and manipulated objects from video data, subsequently applying them to robotic manipulation tasks. Nevertheless, these methods have predominantly targeted robots equipped with simple gripper-based end effectors, failing to generalize effectively to dexterous hands due to the vastly greater complexity of action space. Despite recent advances that utilize kinematic retargeting approaches to produce human-like robotic motions [6, 7, 8, 9, 10], these approaches still fall short in achieving physically-aware pose retargeting and bridging the embodiment gap to derive feasible robot actions capable of successfully accomplishing the intended tasks. *A critical limitation lies in the omission of explicit modeling of interactions between robotic hands and manipulated objects, a fundamental component of manipulation tasks.* Consequently, neglecting these interactions undermines the robot's ability to fully understand and adapt to the dynamics of manipulation scenarios.

Motivated by these challenges, we propose HERMES, a versatile human-to-robot embodied learning framework tailored for mobile bimanual dexterous hand manipulation. HERMES offers the following three advantages: **1. Diverse sources of human motion**: Our framework supports several

---

*Equal Contribution

9th Conference on Robot Learning (CoRL 2025), Seoul, Korea.

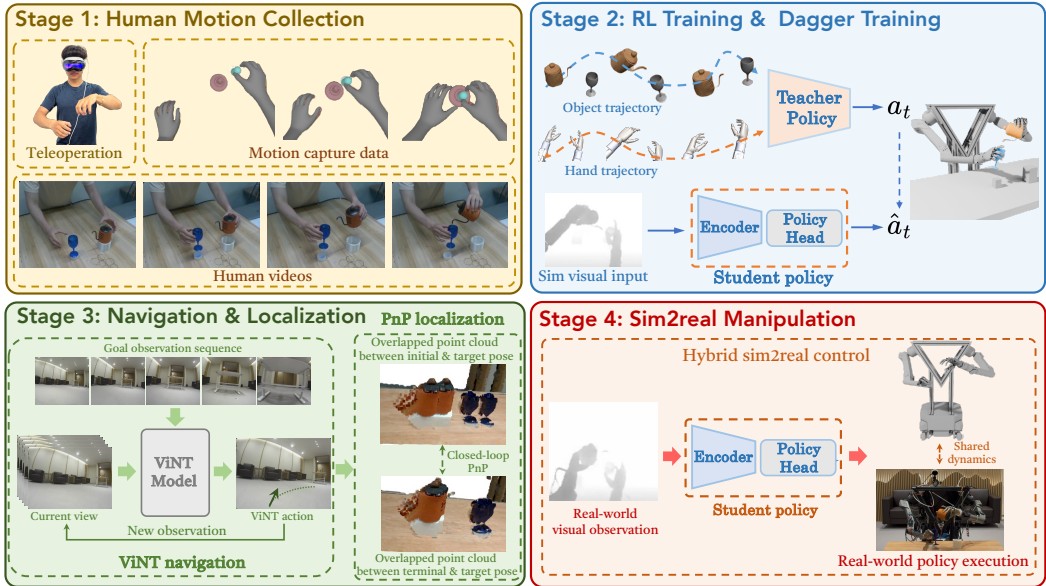

Figure 1: **The main pipeline of HERMES.** HERMES comprises a four-stage pipeline for achieving mobile bimanual dexterous manipulation through sim2real transfer. First, we acquire a one-shot human demonstration drawn from diverse sources. Then, in stage 2, we train a state-based RL teacher policy, then apply DAgger to distill into a vision-based student policy. Following this, HERMES execute long-horizon navigation using ViNT, followed by real-time PnP to finely adjust the robot's pose and achieve precise alignment in stage 3. Once localization is achieved, the student policy is deployed in a zero-shot fashion directly in the real world.

human motion sources, including teleoperated simulation data, motion capture (mocap) data, and raw human videos. We also provide corresponding approaches for data acquisition, enabling HERMES to efficiently transform varied human motion data into robot-feasible behaviors through RL. Furthermore, these tasks share a uniform set of reward terms, obviating the necessity of designing intricate and task-specific reward functions. In contrast to the methods that depend on collecting a large amount of demonstrations, we can achieve generalizable policy by editing a single reference human motion trajectory coupling with RL training. **2. End-to-end vision-based sim2real transfer**: HERMES facilitates robust vision-based sim2real transfer by employing DAgger distillation, which converts state-based expert policies into vision-based student policies. Moreover, we introduce a generalized, object-centric depth image augmentation and hybrid control approach, effectively bridging the perception and dynamic sim2real gap. **3. Mobile manipulation capability**: Our method endows robots with mobile manipulation skills. Building upon ViNT [11], we develop a RGB-D based module for precise localization wherein the task is modeled as a Perspective-n-Point (PnP) problem and addressed through an iterative process. This ensures seamless integration with subsequent manipulation tasks and unlock the policy's capacity to operate autonomously across a broad spectrum of real-world environments.

## 2 Method

### 2.1 Collect One-shot Human Motion

To validate the effectiveness and robustness of HERMES, we employ three distinct sources of human motion: teleoperation in simulation, motion capture data obtained from public datasets, and hand-object poses extracted from raw videos. Moreover, by leveraging merely a single human reference trajectory in conjunction with RL training, we are able to derive the generalizable robot policy without the need for collecting extensive demonstrations. More details can be found in Appendix A.

**Synthesize multiple trajectories:** To obtain a more generalizable policy, we perform the trajectory editing for the one-shot human motion reference by randomizing the object's position and orientation

in a predefined range. The hand and object poses across the augmented trajectories are transformed as follows:

$$\hat{\mathbf{A}}^{\text{pose}}\left[\tau_k\right] = \mathbf{T}^{\text{trans}} \cdot \mathbf{A}^{\text{pose}}\left[\tau_k\right]. \tag{1}$$

For any given frame $k$ in the trajectory $\tau$, we apply a transformation matrix $\mathbf{T}^{\text{trans}}$ to alter its pose, where $\mathbf{A}^{\text{pose}}$ may represent either the object pose or the hand pose. By editing the reference trajectory, we enable spatial generalization from a single human motion demonstration, obviating the need to manually collect large numbers of teleoped demonstrations.

Upon obtaining synthesized object and hand trajectories from various data sources, we initially employ the DexPilot retargeting method [12] to map the captured human hand poses onto corresponding robot hand configurations. Subsequently, reinforcement learning is leveraged to refine and adapt the initialized robot behaviors.

## 2.2 Generalizable Reward Design for Manipulation

Standard reinforcement learning typically relies on hand-crafted reward functions tailored to each specific task. However, designing such complicated reward structures often impedes scalability and usability, particularly for the dexterous hand. To alleviate this issue, we leverage one-shot human demonstration combined with a generalizable reward formulation, enabling the reuse of a unified reward function across tasks and facilitating the straightforward construction of challenging, long-horizon manipulation tasks.

**Object-centric Distance chain:** Capturing the dynamic spatial relationships between the human hands and the object stands as a pivotal factor in enabling the policy to acquire fine-grained hand-object interaction skills. We designate the coordinates of the fingertips and palm of the hand, along with the center of the object's collision mesh, as keypoints. By modeling the temporal evolution of vectors between these keypoints, we formulate the following reward function:

$$r_{\text{chain}} = \begin{cases} \exp\left\{ \frac{1}{n} \sum_{i=1}^{n} \left\| \vec{r}_{\text{ref}}^{(i)} - \vec{r}^{(i)} \right\| \right\}, & \text{if } N_{\text{contact}} \geq N_{\text{num}} \\ 0, & \text{otherwise} \end{cases} \tag{2}$$

where $\vec{r}^{(i)}$ is the vector from object center to the fingertip or palm. Furthermore, we incorporate contact information into this reward term. Specifically, during the computation of the distance chain, we also evaluate the number of contact points between the fingertips and palms of both hand mesh $\mathbf{C}_{\text{hand}}$ and the object's collision mesh $\mathbf{C}_{\text{obj}}$. This reward component is activated only when the number of contact points $N_{\text{contact}}$ exceeds a predefined threshold $N_{\text{num}}$, ensuring that the policy attends to physically meaningful hand-object interactions.

We also incorporate an object trajectory tracking and a power-penalty term to align the policy's behavior with the target object's trajectory and enhance the smoothness of policy execution and to alleviate the jittering actions. We adopt DrM [13], an off-policy method, leverages a dormant ratio mechanism [14] to enhance exploration capabilities and demonstrates high sample efficiency.

## 3 Sim-to-real Transfer

The training of state-based RL policies typically relies on privileged information which is not accessible in real-world deployment scenarios. Consequently, it is imperative to distill the state-based policy into a visual policy for achieving sim2real transfer. We leverage the depth image as visual input. More details can be found in Appendix C.

**DAgger Distillation Training:** In DAgger training, the state-based expert policy acts as the teacher to guide the learning of a visual student policy. In contrast to prior approaches that distill to object masks or segmented images, HERMES directly distills the state into raw visual observations of entire visual scenarios. This design obviates the need for explicit camera calibration and facilitates the acquisition of the robot's in-the-wild generalization capabilities. Furthermore, we introduce a

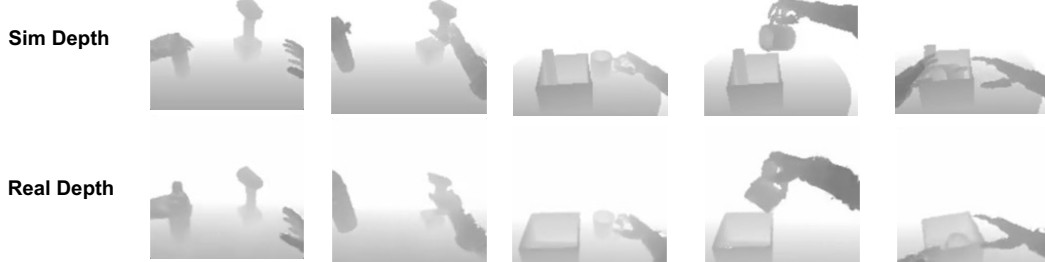

Sim Depth

Real Depth

Figure 2: **Depth image visualization.** After applying our preprocessing pipeline, the depth representations of the hand and object exhibit a strong semantic correspondence, highlighting the efficacy of HERMES in bridging the sim2real gap.

series of auxiliary design choices aimed at enhancing both the asymptotic performance of DAgger training.

**Hybrid sim2real control:** To mitigate the gap between simulation and real-world dynamics as well as proprioceptive information, we adopt a hybrid control strategy: real-world visual observations are used to infer the actual action, which is then applied to the simulation environment to perform a forward step. The updated joint positions of the simulated robot are subsequently transferred to the real robot for execution. By sharing the same Inverse Kinematics (IK) method and dynamic parameters across simulation and the real world, this approach not only enables the policy to adapt its behavior based on real-world environmental variations but also effectively narrows the sim2real discrepancy.

## 4   Navigation Methodology

We choose ViNT [11] for achieving image-goal robotic navigation. ViNT not only enables long-range, in-the-wild navigation but also demonstrates effective zero-shot generalization capability without necessitating model fine-tuning. For our mobile manipulation tasks, moderate discrepancies between the robot's final pose and the target pose can lead to the manipulation policy failing to finish the task. However, ViNT does not guarantee termination within a sufficiently tight error bound. To address this, we introduce a local refinement step after ViNT completes navigation: a closed-loop Perspective-n-Point (PnP) localization algorithm is employed to adjust the robot's pose, ensuring closer alignment with the goal image pose. More details can be found in Appendix B.

## 5   Mobile Manipulation Experiments

To evaluate the mobile manipulation ability of HER-MES, we integrate the entire pipeline across all tasks. Each trained policy is tested over 10 runs. As illustrated in Figure 3, HERMES demonstrates strong real-world navigation, precise localization, and dexterous manipulation capabilities. We also apply the identical manipulation policy equipped with ViNT as a baseline. Figure 3 reveals that, without closed-loop PnP localization, the policy cannot generalize or successfully complete tasks when faced with significant positional and rotational shifts. Conversely, HERMES achieves a notable $+\mathbf{54.0\%}$ improvement in manipulation success rate compared to pure ViNT. These findings underscore that closed-loop PnP localization is the essential bridge linking

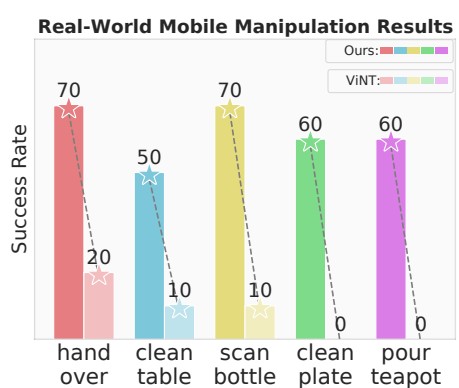

Figure 3: **Real-world mobile manipulation results.**

navigation and manipulation, enabling both modules to synergize for enhanced performance.

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

# Appendix

The visualization of real-world and simulation results are provided in Figure 4 and Figure 5.

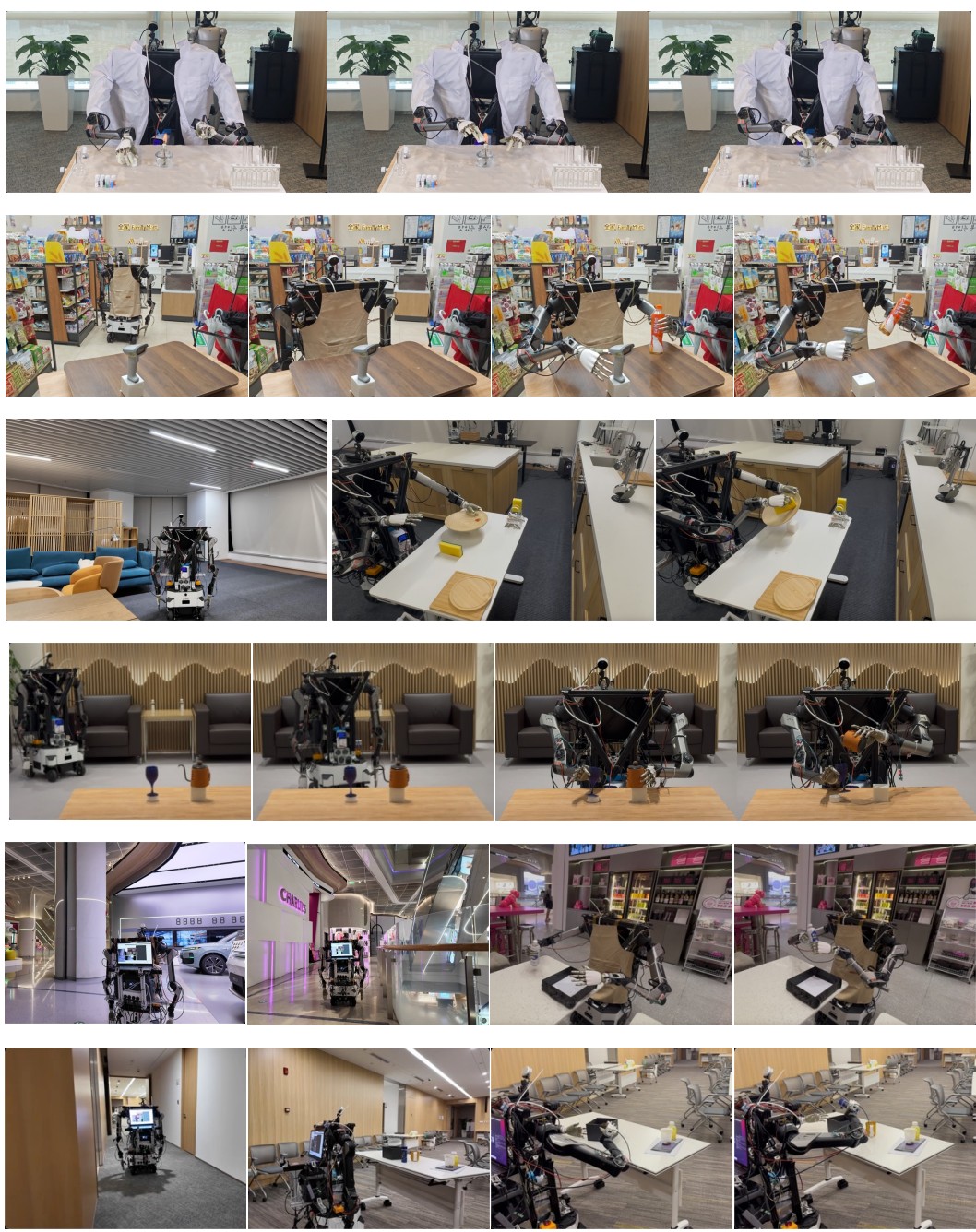

Figure 4: **HERMES exhibits a rich spectrum of mobile bimanual dexterous manipulation skills.** The robot is able to navigate over extended distances in both indoor and outdoor environments, and effectively execute a variety of complex manipulation tasks in unstructured, real-world scenarios, drawing upon behaviors learned from only one-shot human motion.

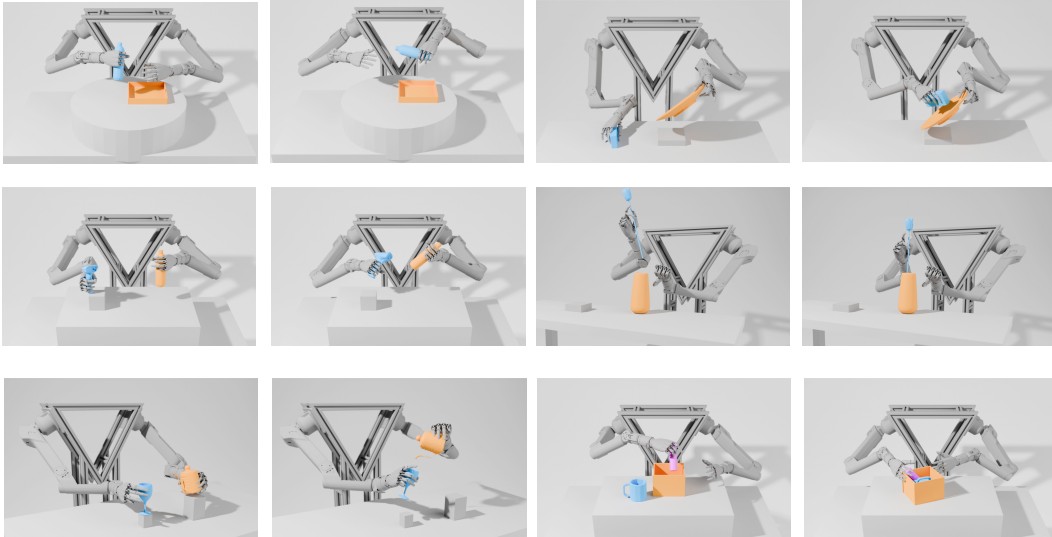

Figure 5: **Simulation training visualization.** We visualize the majority of the training tasks. Leveraging a single reference trajectory in conjunction with a general reward design, HERMES can convert diverse human motion sources into robot feasible behaviors via RL training.

## A  Collect One-shot Human Motion

**Teleoperation in simulation:** We provide access to the pre-configured simulation that enables direct teleoperation of the robot for collecting demonstrations. The Apple Vision Pro is utilized to extract hand poses and arm movements, with data captured at a frequency of 75 Hz.

**Mocap data:** In contrast to direct teleoperation in simulation, retargeting mocap data to robotic hands presents significant challenges due to the embodiment gap between human and robotic hand structures. This discrepancy renders the retargeted trajectories from mocap data unsuitable for direct replay in simulation. Consequently, RL is often employed to enable robots to learn the desired behaviors from reference trajectories. In our study, we utilize the OakInk2 mocap dataset [15] to acquire human motion data for this purpose.

**Extracted arm and hand poses from videos:** Leveraging video data holds considerable promise for unlocking vast quantities of information to facilitate robot learning. To this end, we also provide a pipeline for extracting human hand poses and object trajectories directly from raw

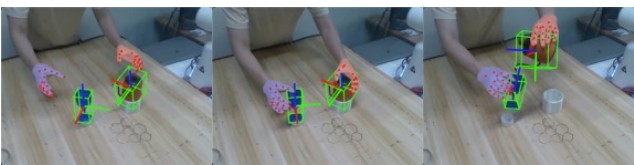

Figure 6: **Pose extraction from videos.**

video. To acquire the hand poses, we first employ WiLoR [16] to detect the hands in each video frame and extract 2D hand keypoints along with their corresponding 3D counterparts. We then select a relatively stable subset of keypoints for the subsequent estimation, specifically those located at the wrist and the metacarpophalangeal joints. The spatial translation of the wrist in the camera coordinate system is estimated by solving a Perspective-n-Point (PnP) problem [17] based on the 2D-3D correspondences, while the palm's orientation is derived by fitting a plane to the selected 3D keypoints. Regarding the manipulated objects, we employ FoundationPose [18] to estimate the object poses directly from video frames, and utilize ARCode [19] scanning to reconstruct the object mesh. By leveraging the aforementioned procedures, we can align the hand and object poses extracted from the video with the robot's frame to facilitate the subsequent learning process.

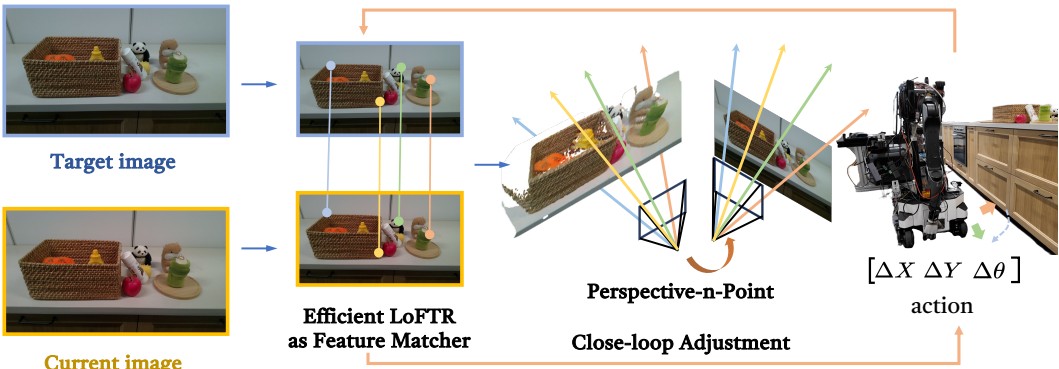

Figure 7: **The pipeline of closed-loop PnP localization.** We first employ the efficient LoFTR to extract dense visual features, followed by estimating the transformation between the current frame and the goal location via solving the PnP problem. Subsequently, we use PID controller to execute the action. This entire process is executed in a closed-loop manner and continues iteratively until the spatial discrepancy between the robot's current pose and the goal falls below a predefined threshold.

## B Navigation Details

As shown in Figure 7, we first utilize the neural feature matching module Efficient LoFTR [20] to detect the correspondence between the current robot captured image $I_c$ and the goal image $I_g$. Then the detected features are lifted to 3D space with respect to the robot's current coordinate frame by leveraging the camera intrinsic matrix and the depth map. Next, we leverage the RANSAC PnP [21] and refine PnP algorithm [22, 23] to compute the relative rotation and translation between the robot's current viewpoint and the goal pose that can minimize the reprojection error. By leveraging real-time feedback from PnP as the robot incrementally converges toward the target pose, we are able to iteratively refine the pose estimation, thereby attaining more accurate visual correspondence. After getting the target pose calculated by our closed-loop PnP localization algorithm, we utilize a Proportional-Integral-Derivative (PID) controller [24] to adjust the pose of our robot. The input of the controller is the instantaneous position and orientation error between the robot's desired state and its actual state.

## C Sim-to-real Transfer

**Leveraging depth image as visual input:** Prior work has explored the use of depth images for vision-based sim2real transfer. However, they often necessitate intricate and highly customized augmentation strategies to bridge the gap. In this work, we introduce a more versatile, manipulation-tailored egocentric depth-image augmentation method. Specifically, we clip depth values beyond a threshold distance **d** (set per task). For real depth images, missing depth values resulting from edge capture failures are filled in with the maximum depth. To emulate real-world edge noise and blur in simulation, we augment simulated depth images by adding Gaussian noise and Gaussian blur during training. Additionally, to mimic missing depth values, we randomly set 0.5% of pixel values in simulation-rendered images to the maximum depth. As illustrated in Figure 2, our augmentation not only semantically aligns simulated renderings with real-world depth images, but also preserves crucial depth disparity cues essential for accurate visuomotor control.

