# OpenReview forum: "HERMES: Human-to-Robot Embodied Learning from Multi-Source Motion Data for Mobile Dexterous Manipulation"
_robot-learning.org/CoRL/2025/Workshop/Dexterous_Manipulation — CoRL 2025 Workshop Dexterous Manipulation Spotlight_

### Official Review · Reviewer_xzrw · 2025-09-08
**Strong performance boost**

**Rating:** 6
**Confidence:** 1

**Review:**

Paper proposes method called HERMES for training robots dexterous manipulation.

Due to the lack of expertise in robotics domain I'm not able to evaluate the paper properly.

I have no major concerns about the content or results, however I'm not aware of related work which is also not covered in the paper.

HERMES proposes framework for training dexterous manipulation from various data sources and strongly improves performance when compared to ViNT. This seems as a great contribution.

The only two disadvantages that I see are:
* The absence of model details and used hyperparameters.
* The lack of ablations on design choices, e.g. importance of each step or choice of novel hyperparameters such as $N_{num}$

---

### Official Review · Reviewer_AoFF · 2025-09-08
**Review for HERMES: Human-to-Robot Embodied Learning from Multi-Source Motion Data for Mobile Dexterous Manipulation**

**Rating:** 7
**Confidence:** 4

**Review:**

The paper introduces a framework for mobile bimanual dexterous manipulation that learns from diverse human motion sources (teleoperation, mocap, videos). Using RL and DAgger, the system converts one-shot human demonstrations into generalizable robot behaviors. A key strength is its robust sim-to-real transfer, achieved via depth-image augmentation and hybrid control, enabling zero-shot real-world deployment. The integration of ViNT-based navigation with closed-loop PnP localization further ensures precise alignment for long-horizon tasks. Experiments show strong real-world success, with over 50% gains compared to baselines. However, the sim-to-real hybrid control feels unnecessarily complicated, and the need for additional real-world fine-tuning in some tasks weakens the zero-shot generalization claim.

---

### Decision · Program_Chairs · 2025-09-18

Accept (Spotlight)